# An Isocaloric Nordic Diet Modulates *RELA* and *TNFRSF1A* Gene Expression in Peripheral Blood Mononuclear Cells in Individuals with Metabolic Syndrome—A SYSDIET Sub-Study

**DOI:** 10.3390/nu11122932

**Published:** 2019-12-03

**Authors:** Stine M. Ulven, Kirsten B. Holven, Amanda Rundblad, Mari C. W. Myhrstad, Lena Leder, Ingrid Dahlman, Vanessa D. de Mello, Ursula Schwab, Carsten Carlberg, Jussi Pihlajamäki, Kjeld Hermansen, Lars O. Dragsted, Ingibjörg Gunnarsdottir, Lieselotte Cloetens, Björn Åkesson, Fredrik Rosqvist, Janne Hukkanen, Karl-Heinz Herzig, Markku J Savolainen, Ulf Risérus, Inga Thorsdottir, Kaisa S Poutanen, Peter Arner, Matti Uusitupa, Marjukka Kolehmainen

**Affiliations:** 1Department of Nutrition, Institute for Basic Medical Sciences, University of Oslo, 0317 Oslo, Norway; k.b.holven@medisin.uio.no (K.B.H.); Amanda.rundblad@medisin.uio.no (A.R.); 2National Advisory Unit for Familial Hypercholesterlemia, Department of Endocrinology, Morbid Obesity and Preventive Medicine, Oslo University Hospital, 0424 Oslo, Norway; 3Department of Nursing and Health Promotion, Faculty of Health Sciences, OsloMet—Oslo Metropolitan University, 0130 Oslo, Norway; mmyhrsta@oslomet.no; 4Mills AS, Sofienberggt. 19, 0558 Oslo, Norway; lena.leder@mills.no; 5Department of Medicine (H7), Karolinska Institute, 17176 Stockholm, Sweden; Ingrid.dahlman@ki.se (I.D.); peter.arner@ki.se (P.A.); 6School of Medicine, Institute of Public Health and Clinical Nutrition, University of Eastern Finland, 70211 Kuopio, Finland; vanessa.laaksonen@uef.fi (V.D.d.M.); ursula.schwab@uef.fi (U.S.); Jussi.Pihlajamaki@uef.fi (J.P.); Kaisa.Poutanen@vtt.fi (K.S.P.); matti.uusitupa@uef.fi (M.U.); marjukka.kolehmainen@uef.fi (M.K.); 7Department of Medicine, Endocrinology and Clinical Nutrition, Kuopio University Hospital, 70029 Kuopio, Finland; 8Institute of Biomedicine, University of Eastern Finland, 70211 Kuopio, Finland; carsten.carlberg@uef.fi; 9Department of Endocrinology and Internal Medicine, Department of Clinical Medicine, Aarhus University Hospital, Aarhus University, 8200 Aarhus, Denmark; kjeld.hermansen@aarhus.rm.dk; 10Department of Nutrition, Exercise and Sports, Faculty of Science, University of Copenhagen, 2200 Copenhagen, Denmark; ldra@nexs.ku.dk; 11Unit for Nutrition Research, University of Iceland and Landspitali—The National University Hospital of Iceland, 101 Reykjavík, Iceland; ingigun@hi.is (I.G.); ingathor@hi.is (I.T.); 12Biomedical Nutrition, Pure and Applied Biochemistry, Lund University, 221 00 Lund, Sweden; Lieselotte.cloetens@tbiokem.lth.se (L.C.); bjorn.akesson@tbiokem.lth.se (B.Å.); 13Department of Clinical Nutrition, Skåne University Hospital, 221 00 Lund, Sweden; 14Department of Public Health and Caring Sciences, Clinical Nutrition and Metabolism, Uppsala University, 751 22 Uppsala, Sweden; Fredrik.rosqvist@pubcare.uu.se (F.R.); ulf.riserus@pubcare.uu.se (U.R.); 15Institute of Clinical Medicine, Department of Internal Medicine and Biocenter Oulu, University of Oulu, Medical Research Center, Oulu University Hospital, 90220 Oulu, Finland; janne.hukkanen@oulu.fi (J.H.); markku.savolainen@oulu.fi (M.J.S.); 16Institute of Biomedicine, Biocenter of Oulu, Medical Research Center, Faculty of Medicine, University of Oulu, and Oulu University Hospital, 90220 Oulu, Finland; karl-heinz.herzig@oulu.fi; 17Department of Gastroenterology and Metabolism, Poznan University of Medical Sciences, 60572 Poznan, Poland; 18VTT Technical Research Centre of Finland, 021100 Espoo, Finland

**Keywords:** metabolic syndrome, randomized controlled dietary intervention, gene expression, peripheral blood mononuclear cells, inflammation

## Abstract

A healthy dietary pattern is associated with a lower risk of metabolic syndrome (MetS) and reduced inflammation. To explore this at the molecular level, we investigated the effect of a Nordic diet (ND) on changes in the gene expression profiles of inflammatory and lipid-related genes in peripheral blood mononuclear cells (PBMCs) of individuals with MetS. We hypothesized that the intake of an ND compared to a control diet (CD) would alter the expression of inflammatory genes and genes involved in lipid metabolism. The individuals with MetS underwent an 18/24-week randomized intervention to compare a ND with a CD. Eighty-eight participants (66% women) were included in this sub-study of the larger SYSDIET study. Fasting PBMCs were collected before and after the intervention and changes in gene expression levels were measured using TaqMan Array Micro Fluidic Cards. Forty-eight pre-determined inflammatory and lipid related gene transcripts were analyzed. The expression level of the gene tumor necrosis factor (TNF) receptor superfamily member 1A *(TNFRSF1A)* was down-regulated (*p* = 0.004), whereas the nuclear factor kappa-light-chain-enhancer of activated B cells (NF-κB) subunit, *RELA*
*proto-oncogene*, was up-regulated (*p* = 0.016) in the ND group compared to the CD group. In conclusion, intake of an ND in individuals with the MetS may affect immune function.

## 1. Introduction 

The metabolic syndrome (MetS) includes a cluster of related risk factors causing increased risk of cardiovascular diseases (CVD) and type 2 diabetes mellitus (T2DM). Central obesity is one of the major factors causing MetS, and metabolic alterations caused by obesity are associated with low-grade chronic inflammation [1,2,3]. The development of MetS is associated with a sedentary lifestyle, excessive energy intake, and an unhealthy diet [4]. It is well known that a Mediterranean-style dietary pattern reduces the risk of MetS [4,5]. The biological mechanisms causing the beneficial effects of a healthy diet are, however, largely unknown.

Peripheral blood mononuclear cells (PBMCs) are immune cells consisting of lymphocytes and monocytes. It is well established that a number of dietary factors modulate gene expression profiles in PBMCs [6,7,8,9,10,11,12,13,14,15]. Since these cells play a key role in the process of inflammation and are exposed to many of the same circulating factors as organs and the arterial wall, they may provide information on how the diet influences systemic inflammation and metabolic changes in peripheral tissues [15].

We have previously shown that a Nordic diet (ND) improved the lipid profile, and the circulating inflammatory marker IL-1 receptor antagonist (IL-1Ra) in individuals with MetS compared to a control diet (CD) (The SYSDIET study) [16]. In addition, we have shown, using global transcriptome profiling, that an ND resulted in the differential expression of inflammatory gene pathways in subcutaneous adipose tissue and PBMCs compared with a CD [12,14]. The ND also resulted in the down-regulation of toll-like receptor 4 (*TRL4*), interleukin 18 (*IL18*), and thrombospondin (*CD36*), and the up-regulation of peroxisome proliferator-activated receptor delta (*PPARD*) after a 2 h oral glucose tolerance test in PBMCs [13].

In this sub-study of the SYSDIET study, our specific aim was to examine the effect of a ND compared to a CD on pre-determined inflammatory and lipid related gene transcripts in fasting PBMC samples. To further understand the mechanisms behind the improved lipid profile and the possible anti-inflammatory effect of ND, we hypothesized that the intake of an ND compared to a CD would alter the expression of inflammatory genes and genes involved in lipid metabolism.

## 2. Materials and Methods

### 2.1. The SYSDIET Study

The SYSDIET study was a randomized controlled multi-center study conducted in 2009–2010 in Kuopio and Oulu (Finland), Uppsala and Lund (Sweden), Aarhus (Denmark) and Reykjavik (Iceland), as previously described [16]. The primary outcome was glucose tolerance and insulin sensitivity. The secondary outcomes were related to MetS risk factors, i.e., blood pressure, serum lipids, inflammatory markers and gene expression. The detailed information on the study design and the main measurements have been described previously [16]. Briefly, after a one-month run-in period, the participants were randomized into a CD group or an ND group for 18 to 24 weeks. The main inclusion criteria were a body mass index (BMI) of 27–38 kg/m^2^, 30–65 years of age, and two other International Diabetes Federation (IDF) criteria for MetS [17]. A stable use of anti-hypertensive and lipid lowering medication during the intervention was allowed. The main exclusion criteria have been described previously [16].

The major visits were in the beginning (0 week), at 12 weeks, and at either 18 or 24 weeks (end of the study). The diets were isocaloric and the study participants were instructed to keep physical activity and body weight constant and their smoking and drinking habits or drug treatment during the study unchanged. All study participants provided written informed consent and local Ethical committees of all the centers included in the current analysis (Research Ethics Committee of the Hospital District of Northern Savo and Northern Ostrobothnia Hospital District, Oulu, Finland and Regional Ethical Review Board, Lund) approved the study protocol in accordance with the Helsinki Declaration. The study is registered at clinicaltrials.gov as NCT00992641.

For this sub-study of SYSDIET, we included participants (n = 94) who had given PBMCs in Kuopio, Lund, and Oulu, and who fulfilled the inclusion criteria (Figure 1). In total, 54 participants in the ND group and 40 in the CD group were included, as previously reported by Leder et al. [13]. The maximum weight change during the study was +/−4 kg, none of the participants used statins, and the high-sensitivity C-reactive protein (hsCRP) was <10 mg/L at baseline and at the end of the intervention, and the BMI was <39 kg/m^2^, as previously reported [13]. Two of the centers had 24 weeks of study length (Kuopio and Lund) and one center had 18 weeks of study length (Oulu) [16].

### 2.2. Diet

The Nordic nutrition recommendations formed the basis for the ND [18], and the mean nutrient intake in the Nordic countries formed the basis for the CD. The main emphasis in the ND group was whole-grain products, abundant use of berries, fruits and vegetables, rapeseed oil, three meals of fish per week, low-fat dairy products and the avoidance of sugar-sweetened products. More details about the diet is described elsewhere [16]. To assess the dietary intake, the participants filled in a 4-day dietary record during the run-in period (baseline intake) and three times during the intervention period.

### 2.3. Biochemical Measurements

All standard laboratory measurements, and anthropometric measurements were performed locally according to the standard operational procedures [16]. The plasma interleukins, plasma tumor necrosis factor receptor II (TNF RII), and serum high molecular weight (HMW) adiponectin were measured using ELISA, as previously described [16].

### 2.4. Sampling of PBMCs and RNA Extraction

The PBMCs were isolated from blood samples collected after overnight fasting (12 h) using cell preparation tubes (CPT) according to the manufacturer’s instructions (Becton, Dickinson and Company, Franklin Lakes, NJ, USA) within 30 to 45 min. All PBMC samples for the RNA analyses were prepared in the same laboratory (Karolinska Institute, Stockholm). The total RNA was extracted using the RNeasy Mini Kit according to the manufacturer’s instructions (Qiagen, Valencia, CA, USA). The RNA integrity was checked using a Bioanalyzer device (Agilent 2100 Bioanalyzer, Agilent Technologies, Santa Clara, CA, USA).

### 2.5. Real-Time Polymerase Chain Reaction RT-qPCR

The RNA from all samples was reverse transcribed by a high-capacity cDNA reverse transcription kit (Applied Biosystems, Foster City, CA, USA). The selection of genes was primarily based on previous dietary intervention studies where the PBMC gene expression of lipid and cholesterol metabolism genes was modulated.

RT-qPCR was performed on an ABI PRISM 7900HT (Applied Biosystems). TaqMan Array Micro Fluidic Cards (Applied Biosystems) were used for RT-qPCR amplification of the gene transcripts. Three samples were excluded due to a low RNA quality, two samples were excluded due to technical problems, and one samples was excluded after quality control. In total, 88 samples were included in the final analyses. ΔCt was calculated as Ct_(reference gene)_ − Ct_(target)_ and the log ratio (ΔΔCt) was calculated as ΔCt_(end of study)_ − ΔCt_(baseline)_. The TATA-binding protein (*TBP*) was selected as the reference gene for normalization.

### 2.6. Statistical Analysis

Gene expression (Ct-values) was normalized using *TBP* as a reference gene, and the change from the baseline to the end of study was calculated as a log ratio (delta Ct (end of study) - delta Ct (baseline)). The difference between the CD and the ND was tested with a linear regression model, adjusted for age, sex and study center. The differences between baseline and end-of-study gene expression within the groups were tested with a paired *t*-test. The correlations between gene expression changes and changes in various biochemical measures and inflammatory markers were analyzed with Spearman’s correlation using the *rcorr* function. *p*-values < 0.05 were considered significant. All statistical analyses were performed in R.

## 3. Results

### 3.1. Baseline Characteristics 

Eighty-eight individuals (n = 48 ND group, n = 40 CD group) were included in the analyses of the present study (Figure 1). Their baseline characteristics are shown in Table 1.

The changes in nutrient intake were in agreement with the results obtained for the whole study population, as previously reported [13]. The polyunsaturated fatty acid (PUFA) intake increased, saturated fatty acid (SFA) intake decreased and the intakes of β-carotene and fiber increased in the ND group compared to the CD group.

### 3.2. Gene Expression Profiling in PBMCs

Compared to the CD group, the expression level of *TNFRSF1A* was significantly down-regulated in the ND group after intervention (*p* = 0.004), whereas the expression level of *RELA* was significantly increased (*p* = 0.016) after the intervention (Figure 2). No other differences in gene expression among inflammatory genes were observed between the groups after the intervention (Figure 3, Appendix A). No altered expression levels of the lipid metabolism-related genes were observed in the ND group compared to the CD group (Figure 4, Appendix A). 

A within-group analysis showed that the between-group differences were mediated by down-regulation of the *TNFRSF1A* gene in the ND group (*p* = 0.037) (Appendix A), whereas the expression level of the *RELA* gene was significantly down-regulated in the CD group (*p* = 0.007) (Appendix A). In addition, there were several within-group changes in both the ND and the CD groups (Appendix A).

### 3.3. Correlation Analysis

We correlated the changes in gene expression of *TNFRSF1A* and *RELA* with changes in several plasma markers related to health, irrespective of group. Whereas the change in the mRNA level of *TNFRSF1A* did not significantly correlate with any of the circulating metabolites, the change in the mRNA level of *RELA* positively correlated with the change in hsCRP concentration and negatively with the change in low-density lipoprotein (LDL)-cholesterol (LDL-C) concentration, respectively (data not shown).

## 4. Discussion

In the present study, we investigated the impact of a ND compared to a CD on inflammation and lipid metabolism-related genes of PBMCs in individuals with MetS participating in a multi-center intervention study for 18/24 weeks. The ND group had an increased expression level of *RELA* and a decreased expression level of *TNFRSF1A* in their isolated PBMCs compared to the CD group. Our data are in line with previous findings [19] where the expression level of several inflammatory genes such as *TNFRSF1A* and *TNFRSF1B* were down-regulated in PBMCs in a diet-induced weight loss study including 34 overweight individuals with abnormal glucose metabolism and MetS. In the present study, the individuals kept a stable body weight, but despite this, the ND down-regulated the gene expression level of *TNFRSF1A*, which supports the key role of diet in the treatment of the MetS. Our data also agree with, and extend our previous findings using a transcriptome-wide approach, demonstrating that pathways regulating the mitochondrial electron transport chain, immune response, and cell cycle in addition to gene transcripts with common motifs for the transcription factors Nuclear respiratory factor 1 (NRF1), NRF2, and nuclear factor kappa-light-chain-enhancer of activated B cells (NF-κB) were down-regulated in the ND group compared to CD [14].

The increased intake of SFA in contrast to PUFA has been shown to increase liver fat and liver enzyme levels [20,21,22]. While the individuals included in the present study were characterized by having MetS, studies have suggested that both MetS and non-alcoholic fatty liver disease (NAFLD) very early in the progression share several common stimulatory mechanisms [23]. Interestingly, the pro-inflammatory TNF superfamily has been suggested to play a key role in the development of NAFLD and subsequently non-alcoholic steatohepatitis (NASH) [24]. Increased gene expression levels of both *TNF* and *TNFR1* have been shown in humans with NASH, thus supporting a role for the TNFR1 pathway in the progression of NASH. Our finding of a reduced gene expression level of *TNFRSF1A* could thus be associated with attenuation of both inflammation and liver fat accumulation. The inflammatory gene expression profile in PBMCs has previously been shown to reflect the immune component of white adipose tissue [25] and liver lipid metabolism [26], further supporting the notion that PBMC gene expression may be useful in providing information about metabolic health in general.

In addition, the gene transcript *RELA* encoding for the p65 NF-κB subunit was significantly up-regulated in the ND group compared to the CD group. This between-group difference was primarily caused by a down-regulation of this gene within the CD group and therefore, this finding may suggest that the ND plays a role in maintaining homeostasis in the immune response. The activation of the NF-κB system seems to be important in promoting liver inflammation [27]; however, the role of RELA (p65) in the development of atherosclerosis is less clear. Nonetheless, RELA has been shown to be a key regulator of the inflammatory response in macrophages [28], where overexpression of RELA in apolipoprotein E knock-out mice led to reduced atherosclerotic lesion size and higher energy expenditure. Thus, the effects of the activation of RELA and the NFκB pathway may vary in different tissues (liver/macrophages) and depend on the inflammatory state of the individuals (chronic/acute). The observed increase in *RELA* gene expression in PBMCs (precursor for macrophages) in the present study in the ND group compared with the CD group in the present study may, thus, be associated with an atheroprotective effect. However, future studies are warranted to further understand the role of an ND in the regulation of RELA, and the consequences of this regulation in immune cells.

None of the gene transcripts encoding proteins related to lipid metabolism were changed by the ND compared to the CD. We have previously shown that exchanging SFA with PUFA reduced total cholesterol and LDL-C by 9% and 11%, respectively, among subjects with slightly elevated cholesterol [29], and changed the PBMC expression levels of several lipid metabolism-related genes [30]. The reason why we did not observe any changes in lipid metabolism-related genes in the present study, may be due to the smaller effect of the ND on the lipid profile as previously shown [16].

Interestingly, when only comparing changes in gene expression levels within the groups, we observed that there are many more genes regulated within the CD group compared to within the ND group. We can speculate that the reason for the larger effect in the CD group may be that volunteers in this kind of study are usually a health-conscious population. If the subjects in the CD group changed to a not-so-healthy diet, a more pronounced effect on the gene expression profiles might be seen. An unhealthy diet may stress the system more than eating a healthier diet (ND group). These findings are also in line with the results from the white adipose tissue whole genome expression profiles in the SYSDIET study [12].

A major strength of the present study is that we were able to use data from a well-designed randomized controlled dietary intervention study, where the changes in gene expression in the ND group were compared to the changes in a CD group. A problem with longitudinal studies using PBMCs is the short (2–3 weeks) life span of the mononuclear cells. Thus, some cells were exposed to the diet longer than other cells, which might influence the results.

## 5. Conclusions

We found an increased expression of *RELA* and a reduced expression level of *TNFRSF1A* in individuals with MetS after the intervention. These data provide further evidence that the consumption of a ND compared to a CD may affect the immune response at the molecular level in PBMCs. Within the preselected genes that were examined, no gene expression changes in lipid metabolism-related genes were observed between the ND and CD, and further studies are needed to confirm if PBMC gene expression profiles may explain the improvement of the lipid profile in circulation observed after intake of a ND.

## Figures and Tables

**Figure 1 nutrients-11-02932-f001:**
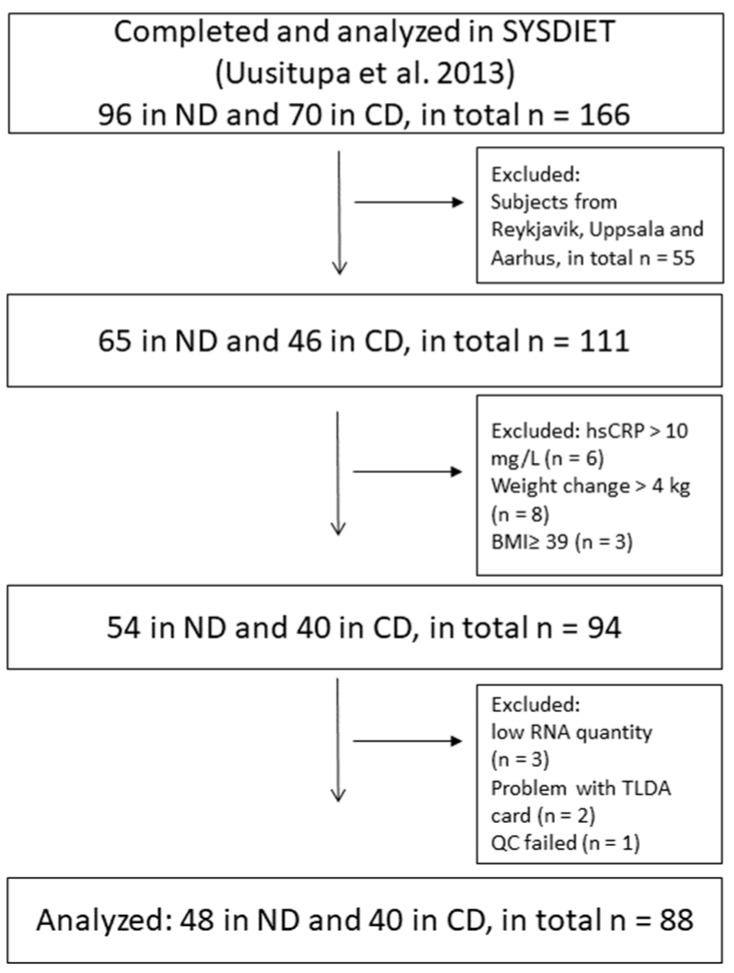
Flow chart of the participants.

**Figure 2 nutrients-11-02932-f002:**
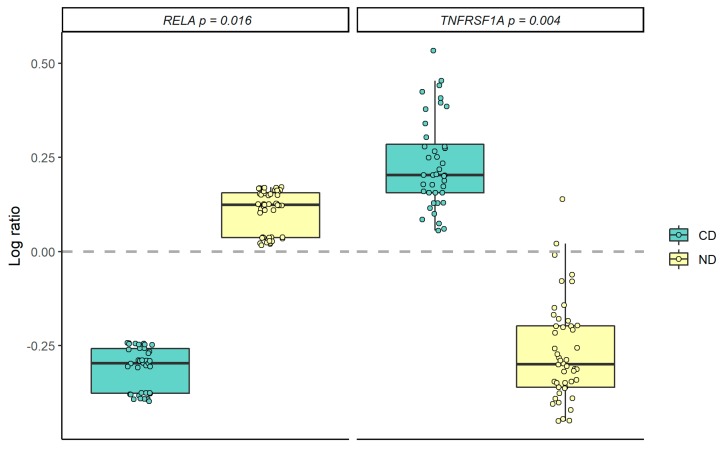
Gene expression changes (log ratio) of *RELA* and *TNFRSF1A* in the Nordic diet (ND) and control diet (CD) groups, adjusted for age, sex and study center. ΔCt was calculated as Ct_(reference gene)_ − Ct_(target)_, and the log ratio (ΔΔCt) was calculated as ΔCt_(end of study)_ − ΔCt_(baseline)_. Differences between the groups were tested with a linear regression model. *p*-values < 0.05 were considered significant.

**Figure 3 nutrients-11-02932-f003:**
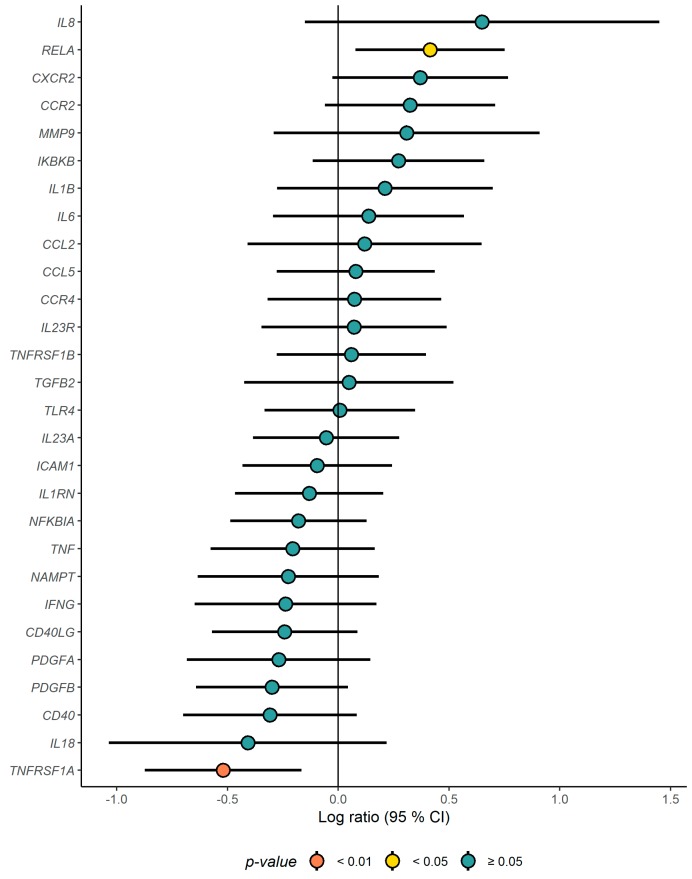
Gene expression changes (log ratio) in the ND relative to the CD of inflammation related genes. ΔCt was calculated as Ct_(reference gene)_ − Ct_(target)_, and the log ratio (ΔΔCt) was calculated as ΔCt_(end of study)_ − ΔCt_(baseline)_. Differences between the groups were tested with a linear regression model, adjusted for age, sex and study center. *p*-values < 0.05 were considered significant.

**Figure 4 nutrients-11-02932-f004:**
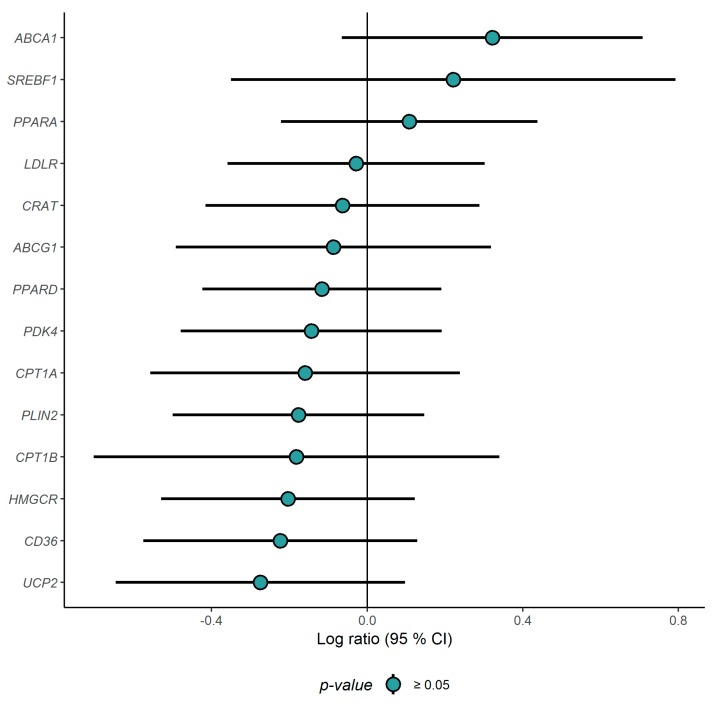
Gene expression changes (log ratio) in the ND relative to the CD of lipid metabolism-related genes. ΔCt was calculated as Ct_(reference gene)_ − Ct_(target)_, and the log ratio (ΔΔCt) was calculated as ΔCt_(end of study)_ − ΔCt_(baseline)_. Differences between the groups were tested with a linear regression model, adjusted for age, sex and study center. *p*-values < 0.05 were considered significant.

**Table 1 nutrients-11-02932-t001:** Baseline characteristics of the participants.

	CD (n = 40)		ND (n = 48)	
Male (n,%)	15	(37.5%)	15	(31.3%)
Age (years)	55.8	(7.8)	54.2	(8.3)
BMI (kg/m^2^)	31.9	(2.7)	31.7	(3.1)
Waist circumference (cm)	105.4	(9.3)	102.6	(9.0)
BP systolic (mmHg)	131	(17)	127	(14)
BP diastolic (mmHg)	82	(12)	83	(10)
Glucose (mmol/L)	5.8	(0.6)	5.9	(0.6)
Insulin (pmol/L)	59.5	(47–80.8)	56.0	(41.8–75.3)
Triglycerides (mmol/L)	1.5	(0.5)	1.5	(0.7)
Total cholesterol (mmol/L)	5.3	(1)	5.3	(1)
HDL cholesterol (mmol/L)	1.3	(0.5)	1.4	(0.3)
LDL cholesterol (mmol/L)	3.3	(0.9)	3.2	(0.9)
hsCRP (mg/L)	1.5	(0.9–3.1)	1.5	(0.8–2.8)
sTNFRII (ng/L)	1900	(415)	1953	(466)
IL-6 (ng/L)	1.3	(1.1–1.7)	1.3	(1–1.8)
IL-10 (ng/L)	0.9	(0.8–1.5)	0.8	(0.8–1.5)
IL-1β (ng/L)	0.12	(0.12–0.17)	0.12	(0.12–0.13)
IL1 Ra (ng/L)	309	(238–463)	301	(220–466)
HMW adiponectin (µg/L)	3.6	(2.2–6.7)	3.9	(2.8–6)

Values are presented either as mean (SD), median (25th–75th percentile) or n (%).

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
