# Peer review of "An Isocaloric Nordic Diet Modulates RELA and TNFRSF1A Gene Expression in Peripheral Blood Mononuclear Cells in Individuals with Metabolic Syndrome—A SYSDIET Sub-Study"

_nutrients, 2019, doi:10.3390/nu11122932_

Round 1

Reviewer 1 Report

The paper substantially is performing all requirements which are being put for examinations of this type. The review paper is well written and in general well organized.

Reviewer 2 Report

The authors responded positively to my suggestions for revision, which I appreciate.

Reviewer 3 Report

I agree we all the changes made by the authors.  However, I have some remaining questions on the data in figure 2 and associated supplemental data that are significant.

Upon re-examination of the supplementary data, I noted that the graphs are shown as deltaCts. Can the authors confirm that by deltaCt they mean: deltaCT=gene of interest Ct - housekeeping Ct? This should be specified in the figure legend or axis title regardless. If so, in the case of TNFRSF1A there is an increase in deltaCt in the CD, indicating a DECREASE in expression, not and increase. Conversely the deltaCt for RELA decreases at the end of the study for the CD, suggesting an INCREASE in expression. This is opposite of what is indicated in figure 2. I am sorry that I did not propose this question in the original review, but I feel this issue must be answered. Am I missunderstanding something?

Furthermore, does the linear regression model used to statistically analyze the data in figure 2 take into account potential differences in bases-line expression levels between the CD and ND groups? As the data is presented as log ratios compared to base-line for each diet, we have no way to tell how the base-lines vary between the groups. Based on the deltaCts presented in the supplemental material, this may be a significant issue for TNFRSF1A. 

Finally, what do the black dots represent in figure 2?

Author Response

Reviewer # 3

I agree we all the changes made by the authors.  However, I have some remaining questions on the data in figure 2 and associated supplemental data that are significant.

1.Upon re-examination of the supplementary data, I noted that the graphs are shown as deltaCts. Can the authors confirm that by deltaCt they mean: deltaCT=gene of interest Ct - housekeeping Ct? This should be specified in the figure legend or axis title regardless. If so, in the case of TNFRSF1A there is an increase in deltaCt in the CD, indicating a DECREASE in expression, not and increase. Conversely the deltaCt for RELA decreases at the end of the study for the CD, suggesting an INCREASE in expression. This is opposite of what is indicated in figure 2. I am sorry that I did not propose this question in the original review, but I feel this issue must be answered. Am I missunderstanding something?

Response: As the reviewer pointed out, in the supplementary figures data are shown as ΔCt. However, ΔCt is calculated as Ct(reference gene) – Ct(target), which is similar to what we did in a  previous paper by Rundblad et al (Rundblad A et al., J Nutr Sci. 2018). We agree that this should be specified in the figure legend as well in the methods section, and this information is  now added in  the revised version. Hence, an increase in ΔCt is an increased expression of the target gene, which is in agreement with figure 2. Also, by calculating ΔCt as Ct(reference gene) – Ct(target), the ΔΔCt (ΔCt(end of study) – ΔCt(baseline)) is the log ratio, as shown in figure 2, without changing the sign of the number.

Page 4 lines 145-146.

2.Furthermore, does the linear regression model used to statistically analyze the data in figure 2 take into account potential differences in bases-line expression levels between the CD and ND groups? As the data is presented as log ratios compared to base-line for each diet, we have no way to tell how the base-lines vary between the groups. Based on the deltaCts presented in the supplemental material, this may be a significant issue for TNFRSF1A. 

Response: We agree with the reviewer that the baseline gene expression may affect the results. However, we chose to perform the statistical analyses in the same manner as we did in a previous SYSDIET paper that examined gene expression with a transcriptomics approach, adjusting for bmi, gender and age (Myhrstad et al., Mol Nutr Food Res. 2019).

3.Finally, what do the black dots represent in figure 2?

Response: We thank the reviewer for pointing out these black dots, so that we are able to remove them. Figure 2 is made up of two layers. Layer 1 is a boxplot, and layer 2 is the individual values fitted to the model. The boxplot, by default, represents data beyond 1.5 IQR from the hinges (outliers) as dots. Thus, the outliers are both plotted by the boxplot layer and the point layer. In the revised version of the manuscript, the black dots are removed to avoid plotting the outliers twice.

New figure 2

This manuscript is a resubmission of an earlier submission. The following is a list of the peer review reports and author responses from that submission.

Round 1

Reviewer 1 Report

This study takes advantage of archived samples from the 2013 SYSDIET samples in order to assess dietary effects of a healthy Nordic diet on PBMC gene expression levels for selected immune modulatory and lipid metabolism genes.

Comments:

Lines 119-120: Did they break down the analysis into the 2 study lengths (which vary by 6 weeks)? Is there an effect of time on the results?

Lines 170-173: In the text when the authors to ND and CD but in Table 1 they use "control" and "SYSDIET", this is confusing. Further, what is the reason for using mean sometimes and median other times within the table?

Lines 187-188: TNFRSF1A shows a strong trend for increase in the CD group (0.06), does this play a significant role in the noted decrease in gene expression in the ND?

Lines 198-190: The statement in the title that a healthy ND regulates RELA is unfounded given that implementation of the ND has no effect on this gene from free feeding baseline levels. The conclusion here is that their CD decreases RELA expression as far as I can tell. The authors have not made it clear to me how this can be interpreted as "The improvement in diet in the ND group led to increased expression level of RELA and decreased expression level of TNFRSF1A in PBMCs 225 compared to the CD." as stated in line 224 of the discussion (though the ND did increase TNRFS1A expression). Indeed the CD induces a change in gene expression from baseline in more genes than does the ND. In the latter, ABCA1 and IL8 increased significantly under the ND, with no changes in the CD; this seems like a potentially significant result on its own but is not discussed.  

Lines 190-191: Why do the authors not discuss the in group changes? Surely the changes in the control group indicate that the control diet modifies gene expression in comparison to free feeding. Given that the diet is meant to mimic this, how relevant is the diet as a control?

Line 204: It is unclear what changes in gene expression the authors are talking about, the changes within groups as reported in the supplemental figures or between groups as reported in figure 2?  How did the plasma markers change within and between the CD and ND groups? Where are the changes in these factors reported? Are the readers of the article expected to refer to the original SYSDIET study (would this be possible since this report is on a sub-group of the original study)? How are they calculated (within groups or between groups?)

Lines 206-207: As per above, I can not see how this should not be interpreted as a negative correlation as the only 'change' observed is a decrease in response to the CD, not an increase in response to ND.

Lines 210-215: None of these genes were reported to be changed between groups and most do not even change within the groups, so what changes are the authors referring to? (minor: note the formatting issues in this section.)

Lines 231-232: Were the genes identified in this study also changed in this previous study? If so, what is novel about the current study?  Does isolating the selected sub-group for study produce a new understanding of the effects of the ND on PBMCs?

Author Response

We thank the reviewers for their thorough and thoughtful evaluation of the manuscript as well as critical and helpful comments.

Reviewer # 1

This study takes advantage of archived samples from the 2013 SYSDIET samples in order to assess dietary effects of a healthy Nordic diet on PBMC gene expression levels for selected immune modulatory and lipid metabolism genes.

Comments:

Lines 119-120: Did they break down the analysis into the 2 study lengths (which vary by 6 weeks)? Is there an effect of time on the results?

Response: We appreciate this comment, and since the length of the study differed in study centers, we adjusted for study center in the linear regression model, which takes the study length into account.

Lines 170-173: In the text when the authors to ND and CD but in Table 1 they use "control" and "SYSDIET", this is confusing. Further, what is the reason for using mean sometimes and median other times within the table?

Response: We apologize for this mistake, and have now corrected Table 1. We prefer to show mean and median values based on if the data are normalized or not.

Lines 187-188: TNFRSF1A shows a strong trend for increase in the CD group (0.06), does this play a significant role in the noted decrease in gene expression in the ND?

Response: it is correct that TNFRSF1A increase in the CD group (P=0.06), but it also decrease significantly in the ND group (P=0.037), and therefore both the slight increase in the CD and the decrease in ND will naturally have an impact the reduction when we compare the groups.

Lines 198-190: The statement in the title that a healthy ND regulates RELA is unfounded given that implementation of the ND has no effect on this gene from free feeding baseline levels. The conclusion here is that their CD decreases RELA expression as far as I can tell. The authors have not made it clear to me how this can be interpreted as "The improvement in diet in the ND group led to increased expression level of RELA and decreased expression level of TNFRSF1A in PBMCs 225 compared to the CD." as stated in line 224 of the discussion (though the ND did increase TNRFS1A expression). Indeed the CD induces a change in gene expression from baseline in more genes than does the ND. In the latter, ABCA1 and IL8 increased significantly under the ND, with no changes in the CD; this seems like a potentially significant result on its own but is not discussed.

Response:We thank the reviewer for letting us clarify our findings. Our focus has been to examine the effect of ND compared to CD. Therefore, we have primarily focused on the changes in gene expression between the groups. We included within analysis since we observed no change in RELA in ND, but this gene was significantly downregulated in CD group (P=0.07). We also find it interesting that there are many more genes regulated within the CD group, compared to within ND group. We have now added this observation in the discussion, and it may be speculated that  the reason for the larger effect of CD in gene expression profiles may be that an unhealthy diet may stress the system more than  eating a more healthier diet. We have also observed that volunters in these kind of studies are usually a health aware population, and thus, their diets might be healthier in the beginning, and when they change to not so healthy diet, the more pronounced effects might be seen in CD group. These findings are also  in line with the paper by Kolehmainen et al., (Am J Clin Nutr. 2015 Jan;101(1):228-39), where white adipose tissue whole genome expression profiles were analysed from the SYSDIET study.

Lines 190-191: Why do the authors not discuss the in group changes? Surely the changes in the control group indicate that the control diet modifies gene expression in comparison to free feeding. Given that the diet is meant to mimic this, how relevant is the diet as a control?

Response: Please see our anwer above. Since the aim was to compare the effect of ND with the CD and thus, we focused on the between group changes.

 Line 204: It is unclear what changes in gene expression the authors are talking about, the changes within groups as reported in the supplemental figures or between groups as reported in figure 2?  How did the plasma markers change within and between the CD and ND groups? Where are the changes in these factors reported? Are the readers of the article expected to refer to the original SYSDIET study (would this be possible since this report is on a sub-group of the original study)? How are they calculated (within groups or between groups?)

Response: When we did the correlations, we included all individuals (n=88) from  the current substudy, not only those in the ND group. We performed the correlations to see whether overall changes in  a gene transcript or a plasma markers were associated to see if we could elucidate further the role of gene expression changes on health markers. However, based on this and the next two comments, we have taken the correlations out of  the manuscript, and we only describe in the text that the change in the mRNA level of TNFRSF1A did not significantly correlate to any of the circulating metabolites, the change in the mRNA level of RELA correlated positively with the change in hsCRP concentration and negatively with the change in low-density lipoprotein (LDL)-cholesterol (LDL-C) concentration, respectively.

Lines 206-207: As per above, I can not see how this should not be interpreted as a negative correlation as the only 'change' observed is a decrease in response to the CD, not an increase in response to ND.

Response: please see our comment above.

Lines 210-215: None of these genes were reported to be changed between groups and most do not even change within the groups, so what changes are the authors referring to? (minor: note the formatting issues in this section.)

Response: We agree witht the reviewers and have taken all correlations analysis out, except for the two significant altered gene transcripts.

Lines 231-232: Were the genes identified in this study also changed in this previous study? If so, what is novel about the current study?  Does isolating the selected sub-group for study produce a new understanding of the effects of the ND on PBMCs?

Response: These genes were not significantly changed in the paper by Myhrstad et al using a global transcriptome approach. We have used two different types of methodology, and also included a larger group of subjects. When we use a global transcriptome approach the main focus is to study changes in metabolic pathways more than identifying single genes per se.  Also the probes used on the microarray may differ from the primer sequence used for the qPCR analysis, which may explain the discrepancy in the results.

Reviewer 2 Report

Nutrients

An isocaloric healthy Nordic…a SYSDIET sub-study

General comments

This manuscript describes the effects of a healthy Nordic diet to a control diet on the expression of inflammatory and lipid related genes in PBMC in subjects with metabolic syndrome. The study was well-designed and generally well-written. The authors would be advised to not describe the Nordic diet as “healthy”, as they did frequently in the text.

Specific comments

Title and elsewhere. Please delete “healthy” before Nordic diet (or ND) as it suggests an implicit bias.

Abstract. Please state the hypothesis for doing this study. What did you expect to find?

52 and elsewhere. Greek symbols are missing from several gene names in my copy (e.g., NF B, and PPAR). 65. “is” related. 87-89. Again, please state the hypothesis of your research. 224. The ND diet likely is an “improvement in diet” but for the sake of neutrality, it probably should be termed a “change in diet”. 235. “healthy ND group suggests that this group did not have MetS. I suggest deleting “healthy”. 236-237. Which liver enzymes were increased by the intake of SFA? 248-253. Because the current study did demonstrate a change in TNF expression, it does not support the conclusion stated in L. 252-252. I recommend deleting this paragraph. 269. “group difference”.

Author Response

Reviewer #2

General comments

This manuscript describes the effects of a healthy Nordic diet to a control diet on the expression of inflammatory and lipid related genes in PBMC in subjects with metabolic syndrome. The study was well-designed and generally well-written. The authors would be advised to not describe the Nordic diet as “healthy”, as they did frequently in the text.

Specific comments

Title and elsewhere. Please delete “healthy” before Nordic diet (or ND) as it suggests an implicit bias.

Response: We appriciate this comment and agree that using the term”healthy” before Nordic diet may implicit bias and we have therefore deleted “healthy”.

Abstract. Please state the hypothesis for doing this study. What did you expect to find?

Response: We apologize for not stating a hypothesis. This is now added, both in  the abstract and in the introduction. To explain the improved lipid profile and the possible anti-inflammatory effect of ND, the hypothesis is that intake of ND compared to CD will alter the expression of inflammatory genes, and genes involved in lipid metabolism.

52 and elsewhere. Greek symbols are missing from several gene names in my copy (e.g., NF B, and PPAR).

Response: This is most probably a formatting mistake, and is now corrected.

“is” related.

Response: This is corrected

87-89. Again, please state the hypothesis of your research.

Response: The hypothesis is; Intake of Nordic diet will alter the expression of inflammatory genes and genes involved in lipid metabolism. This is now added.

The ND diet likely is an “improvement in diet” but for the sake of neutrality, it probably should be termed a “change in diet”.

Response: We appreciate the comment, and have made the change.

“healthy ND group suggests that this group did not have MetS. I suggest deleting “healthy”.

Response: We agree, and have deleted “healthy”

236-237. Which liver enzymes were increased by the intake of SFA?

Response: It was the paper by Rosqvist et al (Rosqvist F, et al. The Journal of clinical endocrinology and metabolism. 2019) that showed that the liver fat accumulation was reflected by plasma alanine aminotransferase (ALT) levels which increased by 18% in the SFA group and remained unchanged in the PUFA group. The between-group difference was significantly different ( P=0.035). We have now stated more clearly that three of the papers showed an effect on liver fat, but only in this paper they measured liver enzymes.

248-253. Because the current study did demonstrate a change in TNF expression, it does not support the conclusion stated in L.

Response:We agree with the referee, and we have therefore taken this part of the discussion out of the paper.

252-252. I recommend deleting this paragraph.

Response: We have deleted this paragraph based on the previous comment.

“group difference”.

Response: This  is corrected.

Reviewer 3 Report

The manuscript by Ulven et al. investigated the effect of a healthy Nordic diet (ND) on changes in gene expression profiles of inflammatory and lipid related genes in peripheral blood mononuclear cells (PBMCs) of individuals with the MetS concluding that intake of ND in MetS subjects may affect the immune function. The results are comprehensive and generally support the authors' conclusions. These findings may benefit from some additional clarification, as detailed below.

Comments

- In table 1 the authors indicate only BMI and not waist circumference, criterion of MetS. Please motivate this decision and revise table 1.

- The authors investigated the impact of a ND on inflammation and lipid metabolism in MetS subjects. Are the results correlated with the number of MetS criteria?

- The manuscript should be edited to correct contextual and layout errors.

Author Response

Reviewer # 3

The manuscript by Ulven et al. investigated the effect of a healthy Nordic diet (ND) on changes in gene expression profiles of inflammatory and lipid related genes in peripheral blood mononuclear cells (PBMCs) of individuals with the MetS concluding that intake of ND in MetS subjects may affect the immune function. The results are comprehensive and generally support the authors' conclusions. These findings may benefit from some additional clarification, as detailed below.

Comments

- In table 1 the authors indicate only BMI and not waist circumference, criterion of MetS. Please motivate this decision and revise table 1.

Response: This is a very valid question, and we have added waist circumference in the table. In general, obese subjects has increased waist circumference above the MetS criteria, and therefore it was not added in the table.

- The authors investigated the impact of a ND on inflammation and lipid metabolism in MetS subjects. Are the results correlated with the number of MetS criteria?

Response: No, this is not done, but it would have been interesting to see if those who fulfill all MetS criteria has different responses compared to those how fulfill only two of the MetS criteria. However, the group size in our study maybe too low to compare these groups separately. The group of subjects are also heterogeneous and grouping the subjects more into those with only lipid dysfunction or only glucose intolerance/insulin resistance would be of great interest for future studies.

- The manuscript should be edited to correct contextual and layout errors.

Response: This is now done, and we apologize for the contextual and layout errors.
